## Simplifying and optimising management of acute malnutrition in children aged 6 to 59 months: study protocol for a community-based individually randomised controlled trial in Kasaï, Democratic Republic of Congo

Cécile Cazes [1], Kevin Phelan [2], Victoire Hubert,[3] Rodrigue Alitanou,[3] Harouna Boubacar,[3] Liévin Izie Bozama,[4] Gilbert Tshibangu Sakubu,[5] Aurélie Beuscart,[1] Cyrille Yao,[6] Delphine Gabillard [1], Moumouni Kinda,[7] Augustin Augier,[2] Xavier Anglaret [1], Susan Shepherd [7], Renaud Becquet [1]

For numbered affiliations see end of article.

**Correspondence to**
Mrs Cécile Cazes;
cecile.cazes@u-bordeaux.fr

## ABSTRACT

**Introduction** Acute malnutrition (AM) is a continuum condition, arbitrarily divided into moderate and severe AM (SAM) categories, funded and managed in separate programmes under different protocols. Optimising acute MAlnutrition (OptiMA) treatment aims to simplify and optimise AM management by treating children with mid-upper arm circumference (MUAC) <125 mm or oedema with one product—ready-to-use therapeutic food—at a gradually tapered dose. Our main objective was to compare the OptiMA strategy with the standard nutritional protocol in children 6–59 months presenting with MUAC <125 mm or oedema without additional complications, as well as in children classified as uncomplicated SAM (ie, MUAC <115 mm or weight-for-height Z-score (WHZ) <−3 or with oedema).

**Methods and analysis** This study was a non-inferiority, individually randomised controlled clinical trial conducted at community level in the Democratic Republic of Congo. Children 6–59 months presenting with MUAC <125 mm or WHZ <−3 or with bipedal oedema and without medical complication were included after signed informed consent in outpatient health facilities. All participants were followed for 6 months. Success in both arms was defined at 6 months post inclusion as being alive, not acutely malnourished per the definition applied at inclusion and without an additional episode of AM throughout the 6-month observation period. Recovery among children with uncomplicated SAM was the main secondary outcome. For the primary objective, 890 participants were needed, and 480 children with SAM were needed for the main secondary objective. We will perform non-inferiority analyses in per-protocol and intention-to-treat basis for both outcomes.

**Ethics and dissemination** Ethics approvals were obtained from the National Health Ethics Committee of the Democratic Republic of Congo and from the Ethics Evaluation Committee of Inserm, the French National Institute for Health and Medical Research (Paris, France).

### Strengths and limitations of this study

► This is the first study of a mid-upperarm circumference (MUAC)-based malnutrition treatment protocol with gradually tapered dose of ready-to-use therapeutic food (RUTF) that uses an individually randomised controlled design.

► The main outcome takes into account sustained health and nutrition status at 6 months after inclusion, including post-recovery relapse and spontaneous recovery for children who are not eligible for RUTF supplementation.

► The methodology used for randomisation (ie, stratification based on WHO definition of severe acute malnutrition (SAM)) allows for two robust analyses: first, how all children with MUAC <125 mm or weight-for-height Z-score <−3 or oedema respond under OptiMA (Optimising acute MAlnutrition); and second, how children only meeting the WHO definition of SAM respond.

► This trial is not a multisite study and only takes place in one country, the Democratic Republic of Congo.

We will submit results for publication to a peer-reviewed journal and disseminate findings in international and national conferences and meetings.

**Trial registration number** NCT03751475. Registered 19 September 2018, https://clinicaltrials.gov/ct2/show/NCT03751475.

## INTRODUCTION

Acute malnutrition (AM) affects an estimated 50 million children under 5 years of age, including 16.6 million severe acute malnutrition (SAM) cases, and is an underlying cause of 800 000 deaths each year worldwide.[1] [2]

Other estimation methods based on incidence instead of prevalence suggest the caseload for this major public health concern may be substantially higher.[3 4] One quarter of all acutely malnourished children are in Africa and as many as 2 million severely wasted children were reported in the Democratic Republic of Congo (DRC) in 2018.[5]

Since the publication of the United Nations (UN) Joint Statement in 2009, millions of children have received treatment for SAM via standard community-based management[6 7] with >80% of children recovering in outpatient care.[8] Despite more than a decade of expanded access to treatment, coverage of nutrition programmes remains remarkably low, as few as 20% of wasted children actually receive treatment[9 10] and the World Bank estimates that it would take an additional US$6.3 billion per year to reach a coverage of 80%.[11] There is, therefore an urgent need to better allocate available resources through the simplification and optimisation of current AM protocols. Adapting programming in three key aspects of case management may allow improved access to treatment for acutely malnourished children.

First, AM is as a continuum condition, but it is arbitrarily divided into moderate acute malnutrition (MAM) and SAM categories. This distinction results in separate programmes overseen by different UN agencies and using different protocols and products—ready-to-use supplementary food (RUSF) or fortified-blended flours for children with MAM and ready-to-use therapeutic food (RUTF) for children with SAM. This complicates supply chain, delivery of care, data collection, and creates confusion and extra work for caregivers and health workers alike.[12] In practice, only SAM treatment is often available, whereas identification and treatment of children earlier in the wasting process would lead to fewer hospitalisations and deaths. The treatment of MAM has been shown to reduce mortality risk by more than 10%.[13]

Second, current case definitions to determine programme eligibility are unnecessarily complicated, using two independent anthropometric criteria: mid-upper arm circumference (MUAC) or weight-for-height Z-score (WHZ). WHZ alone or in combination with MUAC does not offer a clear advantage for identifying children at near-term risk of death over MUAC alone.[14] MUAC-only programming is expanding as evidence accumulates that weight gain and MUAC gain track each other and respond to treatment in similar ways.[15–17] MUAC is, therefore, becoming a stand-alone practical tool for all phases of nutrition programming: screening, admission, monitoring recovery and determining discharge. Furthermore, the simplicity of MUAC bracelets makes it possible to screen children not only in the community but also at home.[18]

Third, therapeutic food rations could be allocated more efficiently. The RUTF ration for treating SAM (130–200 kcal/kg/day) is paradoxical. The amount of RUTF prescribed remains constant or even increases with weight gain up until discharge (ie, as the child recovers from AM), even though weight and MUAC gain is maximal during the first 2–3 weeks of supplementation.[19] Consequently, gradual dose reduction seems to be a more rational use of RUTF which accounts for a large proportion of total malnutrition programme costs.[20]

Results on integrating SAM and MAM treatment in a single MUAC-based protocol using only RUTF at a gradually reduced dose were first published in 2015. This cluster-randomised trial conducted in Sierra Leone found high recovery rates, a reduced caseload of children with SAM, a better coverage rate and reduced RUTF costs when compared with the standard protocol.[21] More recently, a non-inferiority randomised trial in Burkina Faso showed that the reduction of the RUTF dose after the first 2 weeks results in similar weight gain velocity and recovery rates as with the standard dose given throughout SAM treatment.[22]

Our research consortium developed the Optimising treatment for Acute Malnutrition (OptiMA) strategy that simplifies the definition of AM to MUAC <125 mm or the presence of bipedal oedema, and uses a single product for treatment—RUTF—at a gradually reduced dose based on a child's weight and MUAC status (see table 1). The OptiMA strategy was first implemented in a proof-of-concept single-arm trial in Burkina Faso and showed a recovery rate that exceeded national and international Sphere[23] standards for both SAM and MAM (86.3%; 95% CI 85.4% to 87.2%) with excellent health worker adherence to the new RUTF dosage table.[24] However, recovery of children admitted at MUAC <115 mm was poorer than anticipated (70.4%; 95% CI 67.5% to 73.5%) which necessitated a randomised controlled trial to compare OptiMA to current national malnutrition protocols. We therefore planned this individually randomised non-inferiority clinical trial to compare the OptiMA strategy to the DRC nutritional standard protocol in children 6–59 m with uncomplicated AM.

We hypothesised that the OptiMA strategy would be as effective as the current DRC national protocol currently in use, in children 6 to 59 years old as judged by the success rate in the treatment of uncomplicated SAM and MAM at 6 months postrandomisation and in terms of recovery rate in the treatment of uncomplicated SAM.

## Objectives

The principal objective aims to determine, 6 months after inclusion, whether the OptiMA strategy leads a success rate that is non-inferior to that of the standard DRC protocol in use in the same outpatient health facilities. The definition of 'success' is described in the Study outcomes section.

The main secondary objective is to determine whether the recovery rate of children with uncomplicated SAM according to the current WHO definition[25] (ie, MUAC <115 mm or WHZ <–3 or bilateral oedema) managed under the OptiMA protocol is non-inferior to that of the national standard protocol, anticipating that such children treated under OptiMA will receive a smaller overall

**Table 1** Wasting definition, treatment products, calculation of dosage and recovery definition in the DRC national and OptiMA protocol

| | National DRC protocol | | OptiMA protocol | | |
| --- | --- | --- | --- | --- | --- |
| | **SAM** | **MAM** | **Acute malnutrition** | | |
| Wasting definition | MUAC <115 mm or WHZ <−3 or bipedal oedema | MUAC (115–124 mm) or −3 ≤WHZ <−2 | MUAC <125 mm or bipedal oedema | | |
| Treatment product | RUTF 150–200 kcal/kg/day | Super cereal plus 200 g/day (~1000 kcal/day) or RUSF, one 92 g sachet/day (500 kcal/day) | MUAC <115 mm or bipedal oedema | MUAC (115–119 mm) | MUAC (120–124 mm) |
| | | | RUTF 170–200 (kcal/kg/day) | RUTF 125–190 (kcal/kg/day) | RUTF 50–166 (kcal/kg/day) |
| Calculation of dosage | According to the weight | Fixed amount, regardless of weight or MUAC status | According to MUAC status and weight | | |
| Recovery definition | MUAC ≥125 mm or WHZ ≥−1.5 (for 2 consecutive weeks) | MUAC ≥125 mm or WHZ ≥−1.5 If after recovery from SAM: MUAC ≥125 mm and WHZ ≥−1.5 and discharge after 3 months | MUAC >125 mm for 2 consecutive weeks | | |
| | And no oedema for 2 consecutive weeks | And no oedema for 2 consecutive weeks | And No oedema for 2 consecutive weeks | | |
| | | | And minimum 4 weeks in programme and good clinical health | | |

DRC, Democratic Republic of Congo; MAM, moderate acute malnutrition; MUAC, mid-upper arm circumference; OptiMA, Optimising acute MAlnutrition; RUSF, ready-to-use supplementary food; RUTF, ready-to-use therapeutic food ; SAM, sever acute malnutrition; WHZ, weight-for-height Z-score.

RUTF ration than those treated under the DRC national protocol.

Other secondary objectives include description and comparison between the two study arms of the following outcomes: RUTF consumption, relapse after nutritional recovery, outcomes compared with international Sphere references, recovery and success in children who present with both wasting and stunting at inclusion, and to describe the nutritional and clinical status of children hospitalised while enrolled in the study.

## METHODS
This protocol follows the Standard Protocol Items: recommendations for Interventional Trials (SPIRIT) guidelines.[26]

### Design
This study is a non-inferiority individually randomised controlled clinical trial conducted at health centres and in the community. Participants were randomly assigned to either the OptiMA strategy arm (intervention=OptiMA) or the national standard protocol arm (control=Standard).

### Study setting
The trial took place in the Kamuesha health zone, Kasai province in DRC and was nested within a medical and nutritional emergency humanitarian project launched by Alliance for International Medical Action (ALIMA) in

May 2018. The Kamuesha health zone is located in a tropical forest and is particularly isolated.

ALIMA supports the DRC Ministry of Health (MoH) in 9 of 26 health areas in the Kamuesha health zone to address a nutritional crisis occurring after a 2 years of armed conflict with significant population displacement.[27] During the trial preparation phase, we visited all health areas covered by ALIMA and selected four as research sites based on demographic, epidemiological and logistical criteria in addition to observations provided by local stakeholders. According to projections from the last census, the four health areas chosen covered a population of 65 000 people, including 12 000 children aged 6 to 59 months, spread over 60 villages. Health facilities included in the study were the four community-based primary health centres of these four health areas in addition to the general hospital of Kamuesha district which was used for referring participants with medical complications. Most (n=49) villages were less than 10 km away from the primary health centres while 11 were more than 15 km away.

### Eligibility criteria
Children aged 6–59 months, residing in one of the four health areas and meeting at least one of the three following AM criteria were eligible for inclusion in the trial: MUAC <125 mm and/or bilateral pitting oedema (+/++) and/or WHZ <−3. Children with a medical

complication requiring hospitalisation, no appetite, oedema grade +++, with a known allergy to milk, peanuts or RUTFs, with a known chronic pathology (such as sickle cell anaemia, trisomy 21, congenital heart disease, neurological disease), or already enrolled in a malnutrition programme were excluded from the trial. Two categories of children were included in the trial but not eligible for randomisation for ethical concerns. First, children with WHZ <−3 and MUAC ≥125 mm without oedema were systematically included in the control arm, as they were not RUTF eligible according to the wasting definition of the OptiMA strategy. Second, children with a sibling already enrolled in the study were systematically allocated to the study arm of the index sibling.

## Study outcomes

The primary outcome will be judged by a binary composite indicator. Children classified as 'success' fulfil all of the following criteria: alive, not acutely malnourished per the definition applied at inclusion and not having an additional episode of AM throughout the 6-month observation period. All other children are classified as 'unsuccessful'.

The main secondary outcome will be determined among children in both arms of the trial who meet the current WHO definition of SAM at inclusion. For this subgroup, recovery is defined in both arms, after a 4-week minimum duration of treatment as, clinically well (an axillary temperature <37.5°C), absence of bipedal oedema and an MUAC ≥125 mm for the OptiMA arm or MUAC ≥125 mm or WHZ ≥−1.5 for the standard arm, for two consecutive weeks. Additional secondary outcomes are listed in table 2.

## Sample size considerations

We determined the sample size for the principal objective using an expected success rate of 55% and for the secondary main objective an expected recovery rate of 85%. Both of these hypotheses are based on recovery, death, default and relapse rates available in the literature for severely and moderately wasted children supplemented with RUTF and/or RUSF in DRC and other African contexts.[28 29] We also considered spontaneous recovery rate documented for MAM children without treatment in the standard arm.[30] During the preparation phase, we twice visited the four sites included in the study and found that the standard MAM treatment programme was either non-functional or partially functional.

We assumed a margin of non-inferiority of 10%, the same as the three other non-inferiority trials assessing simplified treatment strategies with RUTF for wasted children.[31–33]

To demonstrate non-inferiority of the OptiMA strategy compared with standard protocol, 890 and 490 participants were required for the principal and main secondary objectives, respectively, with 80% power at a unilateral significance level of 2.5% and with an inflation of 15% to account for unexploitable data. We estimated 15% of incomplete data in the light of the challenging reality of

this study setting where families are regularly confronted with barriers to receiving healthcare, insecurity and displacement.

## Randomisation

The trial statistician established and maintained a confidential randomisation list. Randomisation was done in blocks and was stratified at the study site by the severity of AM according to the WHO definition. This double stratification allows for recruitment of comparable groups of SAM and non-SAM children and for simultaneous randomisation at the four sites. The trial data scientist integrated the list into a randomisation software developed for the trial that was programmed on four tablet devices. This software assigned a treatment arm by sequentially drawing from this list each time a randomisation procedure was completed. Once the sample size for the primary objective was reached, only WHO-SAM cases continued to be enrolled and randomised until the sample size required for the main secondary objective was attained.

## Patient and public involvement

To ensure that the trial was understood, accepted and implemented in accordance with local customs and practices, meetings to explain trial eligibility and procedures were organised with community representatives in each village and with nutrition actors prior to recruitment. During the trial's active phase, regular oral presentations were held at the local level. We intend to disseminate the findings through meetings with community representatives and national nutrition actors.

## Enrolment and informed consent procedure

Prior to enrolment, all community health workers, nurse research officers and MoH medical staff implicated in the study were trained on the trial protocol, standardised operating procedures and the correct use of tools developed for data collection. Nurse research officers were specifically trained to enrol and randomise study participants, and monitor home visits and outpatient visits at the health centre. Children were recruited through monthly active mass malnutrition screenings in each of the 60 villages. Concomitantly, children attending outpatient consultations at the four study health centres underwent routine passive malnutrition screening and were enrolled in the trial if eligible. In both active and passive screening, a nurse research officer, assisted by a community health worker, collected anthropometric (weight, height, MUAC, WHZ) and clinical (presence of oedema) information on eligible children. Children then underwent a medical consultation to determine whether they met the criteria for study inclusion. A standard operating procedure (see online supplemental file 1) for assessing child anthropometry (weight, height, MUAC and oedema) were observed in each site. For each eligible child, the caretaker was given a study information sheet detailing the study's aims, treatment protocols, study duration, and

**Table 2** Primary and secondary outcomes in the OptiMA-DRC trial

| | Measurement variable | Denominator | Method of aggregation | Timepoint |
|---|---|---|---|---|
| **Primary** | | | | |
| Success | Composite: alive, not acutely malnourished per the definition applied at inclusion and no additional episode of acute malnutrition throughout the 6-month observation period | Children with MUAC <125 mm or WHZ <−3 or bipedal oedema at inclusion | Proportion | 6 months post-inclusion |
| **Main secondary** | | | | |
| Recovery | Composite: in both arms, a 4-week minimum duration of RUTF treatment, an axillary temperature <37.5°C, absence of bipedal oedema, and for the OptiMA arm a MUAC ≥125 mm and for the standard arm MUAC ≥125 mm or WHZ ≥−1.5, for two consecutives weeks | Children with MUAC <115 mm or WHZ <−3 or bipedal oedema at inclusion | Proportion | Throughout the 6 months needed for study completion after RUTF treatment has been initiated at inclusion |
| **Secondary** | | | | |
| Consumption of RUTF by children with acute malnutrition at inclusion | Sachets of RUTF consumed | Children with MUAC <125 mm or WHZ <−3 or bipedal oedema at inclusion | Median/mean | 6 months post-inclusion |
| Consumption of RUTF by children who recovered from SAM | Sachets of RUTF consumed | Children with MUAC <115 mm or WHZ <−3 or bipedal oedema at inclusion who recovered | Median/mean | At the visit when recovery is determined |
| Total weight gain and daily weight gain in children who recovered from SAM | Total weight gain (g) and weight gain in g/kg/day | Children with MUAC <115 mm or WHZ <−3 or bipedal oedema at inclusion who recovered | Median/mean | At the visit when recovery is determined |
| Total MUAC gain and daily MUAC gain in children who recovered from SAM | Total MUAC gain in mm and MUAC gain in mm/day | Children with MUAC <115 mm or WHZ <−3 or bipedal oedema at inclusion who recovered | Median/mean | At the visit when recovery is determined |
| Total length of RUTF treatment in children who recovered from SAM | Total number of days with RUTF treatment | Children with MUAC <115 mm or WHZ <−3 or bipedal oedema at inclusion who recovered | Median/mean | At the visit when recovery is determined |
| Non-response in children with SAM | Absence of recovery status | Children with MUAC <115 mm or WHZ <−3 or bipedal oedema at inclusion | Proportion | After 12 and 16 weeks of nutritional follow-up of SAM episode at inclusion |
| Relapse to a new episode of SAM (WHO definition) | Children with MUAC<115 or WHZ <−3 or oedema after RUTF treatment | Children with MUAC <115 mm or WHZ <−3 or bipedal oedema at inclusion recovered | Proportion | During 3 months following recovery from SAM episode at inclusion |

MUAC, mid-upper arm circumference; OptiMA, optimizing acute malnutrition; RUTF, ready-to-use therapeutic food; SAM, severe acute malnutrition; WHZ, weight-for-height Z-score.

frequency of clinic visits and community follow-up. This information was explained orally in Tshiluba (local language). Caretakers who agreed to participate in the study indicated their consent by signing (signature or fingerprint) a written consent form. When caretakers were not able to read or write, an impartial witness oversaw the consent process and attested to the caretaker's consent by signing the consent form on her/his behalf. All caretakers were informed about their right to withdraw from the study at any time without affecting the quality of medical care provided to their children. All medical care and nutritional treatment were provided free of charge, regardless of participation in the study.

### Intervention and follow-up
In the intervention arm, all children with a MUAC <125 mm or oedema were supplemented with RUTF, according to the OptiMA dosage table based on the progression of MUAC and weight gain during recovery (ie, RUTF dosage is gradually reduced as a child's weight and MUAC increases) (see table 1). The OptiMA RUTF dosage table provides for a daily caloric intake of 170–200, 125–190 and 50–166 kcal/kg/day in children with a MUAC <115 mm, between 115 and 119 mm and >119 mm, respectively. Children with oedema received the same RUTF ration as children with MUAC <115 mm until oedema resolved, at which point their ration was determined by MUAC and weight (see online supplemental file 2).

In the control arm, children included with MUAC <115 mm and/or WHZ <−3 and/or nutritional oedema were treated with RUTF, according to the DRC national protocol dosage table which is based on a child's weight at each visit (ie, RUTF dosage gradually increases as a child's weight increases). The DRC protocol provides a RUTF daily caloric intake of 150–200 kcal/kg/day during the entire treatment course until recovery in children with SAM and one sachet of RUSF per day (500 kcal/day) in children with MAM.

The differences between the OptiMA and the DRC national standard protocols, summarised in table 1, concern treatment eligibility, definitions of recovery and relapse, the type of therapeutic food provided and RUTF dosage calculation. All other aspects of the standard nutrition protocol were applied to all children in both arms. All cases of SAM according to the WHO definition received amoxicillin 50–100 mg/kg/day for 7 days, and all children received vitamin A and deworming regardless of anthropometry. In both arms, a malaria rapid test was systematically performed at inclusion and at any point during follow-up when a child presented with clinical signs of malaria. If positive, an artemisinin combination treatment was prescribed for the child. At the time of the study, the HIV national programme was not functional in the Kamuesha health zone, hence children with SAM were not offered HIV counselling and testing, as is recommended.[25]

All participants were monitored for 6 months after inclusion (see figure 1). While in the treatment phase, children came to the health centres once a week for outpatient consultations (or every 2 weeks for those children living in villages more than 14 km from the health centre). On discharge from treatment, or immediately after inclusion for children with MAM who did not receive RUTF in the standard arm, children were consulted every 2 weeks in their villages until 6 months post-inclusion was completed. During village visits, a nurse research officer assisted by one or two community health workers monitored the anthropometric and clinical status of these children, referring any child who needed nutritional or medical care to either the primary healthcare facility or to the Kamuesha general hospital.

### Blinding
This was an unblinded trial. Caretakers and study personnel were aware of study arm allocation for participating children due to differences in anthropometric criteria and RUTF ration. On inclusion, the nurse research officer gave each trial participant's caretaker a card specifying the nutritional strategy being followed by the child. Caretakers were asked to present this card in the event of an unplanned outpatient consultation or hospital admission during the study period. In this way, nurses or doctors at any health facility were able to identify the nutritional treatment being followed by the child.

### Data collection and management
After obtaining individual informed consent, all sociodemographic characteristics, anthropometric and clinical data collected during follow-up were documented in the case report forms by the research officers and the healthcare providers trained specifically for this study. All data entered into the database were securely transferred and stored on a server hosted by PAC-CI, a satellite for the French National Agency for Research on AIDS and Viral Hepatitis in Abidjan, Côte d'Ivoire which served as the methodology and coordinating centre for this study. All data and any adverse events, including deaths and hospitalisations, were rigorously monitored both on-site and remotely, according to the data monitoring plan. To ensure patient safety and data integrity, the methodology and coordinating centre continuously supervised data management activities, and the senior scientific project leader and investigators performed field visits on a regular basis. If children failed to attend follow-up visits, the community health worker living in the village visited the household to check on the child, to determine the reasons for absence and to encourage the caretakers to adhere to the planned health centre visit schedule.

### Statistical analyses
All statistical analyses will be performed using RStudio Software (Boston, Massachusetts, USA) as specified by the data analysis plan published on ClinicalTrials.gov prior to performing any analysis. Statistical tests will be

| | Enrolment | Allocation | Post-allocation | | | | | | Close-out |
|---|---|---|---|---|---|---|---|---|---|
| **STUDY PERIOD** | | | | | | | | | |
| TIMEPOINT | | d0 | $m_1$ | $m_2$ | $m_3$ | $m_4$ | $m_5$ | $m_6$ | $m_6$ |
| **ENROLMENT:** | | | | | | | | | |
| **Eligibility screening*** | X | | | | | | | | |
| | | Outpatient visit | | | | | | | |
| **Informed consent** | | X | | | | | | | |
| **Allocation** | | X | | | | | | | |
| **INTERVENTIONS:** | | X | Outpatient** or home visits *** | | | | | | |
| **Standard strategy** | | X | X | X | X | X | X | X | |
| **OptiMA strategy** | | X | X | X | X | X | X | X | |
| **ASSESSMENTS:** | | | | | | | | | |
| **Clinical examination** | | X | X | X | X | X | X | X | X |
| **Anthropometry measurements** | X | X | X | X | X | X | X | X | X |
| | | | Inpatient wards | | | | | | |
| **Hospitalisation if required** | | | X | X | X | X | X | X | X |

**Figure 1** Schedule of enrolment, interventions and assessments OptiMA-DRC overview. *Monthly active screening in 60 villages and passive screening during outpatient visit in four health centres. **Weekly (for those living in villages at 14 km or less from the health centre) or bimonthly (for those living in villages more than 14 km from the health centre) outpatient visits at health centre for participants with ready-to-use therapeutic food (RUTF) supplementation. ***Bimonthly home visits for children without RUTF supplementation. d, day; DRC, Democratic Republic of Congo; m, month; OptiMA, optimising acute malnutrition.

carried out bilaterally with an alpha risk of 5% and unilaterally with an alpha risk of 2.5% for the non-inferiority analysis. Qualitative variables will be described in terms of numbers, percentages and provided with CIs when relevant. If necessary, comparisons of qualitative variables will be made using $\chi^2$ or Fisher exact tests. Quantitative variables will be described in terms of mean, SD and CI or median, range and IQR. If necessary, comparisons of quantitative variables will be made using Student's, Wilcoxon's or the Kruskal-Wallis tests, depending on the distribution of the variable of interest. Time delay type variables will be described in terms of the incidence of occurrence, and the probability of occurrence over time, estimated by the Kaplan-Meier method. If necessary, probability comparisons will be made using log-rank tests, or proportional risk models, after verification of the assumption of proportionality of risks.

### Primary analysis of the primary and secondary main outcomes

Descriptive summaries of participant characteristics by arm at inclusion will be tabulated. Participants included in the main analyses will be described according to the diagram defined by CONSORT (Consolidated Standards of Reporting Trials) recommendations.[34] The occurrence of the primary 'success' and the secondary main outcome 'recovery' will be compared between the two randomisation arms. These two comparisons will be performed both by 'intention-to-treat' basis and on a 'per-protocol' basis. Non-inferiority analyses will be performed on these two outcomes. Only if non-inferiority for these two outcomes is demonstrated will the other secondary analyses will be performed, and, if appropriate, as a superiority analysis. Superiority will be particularly sought for cost outcomes related to RUTF cost-efficiency.

### Sensitivity analysis of the primary outcome

First, rates of 'success' and 'recovery' will be analysed based on all available data. Then a sensitivity analysis will be performed using the maximum bias method. In this analysis, deceased participants, those who have withdrawn their consent, who have been transferred to another health facility or have been lost to follow-up, will be considered to have systematically failed to the allocated treatment.

### Safety

The trial was monitored monthly by an international steering committee and quarterly by a national steering committee where any adverse events were presented. No interim analysis was planned before the final analyses. The data safety monitoring board could have requested

an interim analysis if it deemed it necessary for patient safety.

A child in intervention arm could be removed from the study for two reasons: (1) if he or she had two episodes of >5% weight loss during the course of treatment and (2) if he or she was not SAM at inclusion but deteriorated to SAM during RUTF supplementation.

### Ethics and dissemination

The study was conducted in accordance with the Declaration of Helsinki. We obtained ethical approval with annual renewal from DRC National Ethics Committee (CNES) (94/CNES/BN/PMMF/2018) and from the Ethics Evaluation Committee of the French National Institute for Health and Medical Research (Inserm) (18-545). The final version of the protocol, version 3.1, dated 22 October 2019, is available for sharing on request and included three minor amendments to protocol version 2.0 (dated 6 March 2019) and version 3.0 (dated 18 April 2019) which was shared with national and international ethical committees and did not require a new authorisation. In November 2019, the CNES performed an audit at each site and then renewed approval (152/CNES/BN/PMMF/2019) for the continuation of the trial. All data were anonymised when entered into the database using unique identification numbers. We intend to disseminate the results regardless of positive or negative findings broadly via peer-reviewed publication, conferences and clinical networks targeting academics, policy-makers, clinicians and caregivers.

### DISCUSSION

This trial was anchored in the following foundational principle: the need to merge MAM and SAM nutritional care into a single programme using one anthropometric admission criterion and one type of therapeutic food to simplify care for families, health workers and programme managers. This study will contribute to a developing evidence-base composed of similar studies already published or currently in progress in Sierra Leone, Burkina Faso, South Sudan and Kenya.[31 35] Given that the risk profile associated with low anthropometry is highly contextual, it is important to conduct similar trials in multiple settings. This will be the first such trial in DRC, where we expect a higher proportion of oedematous malnutrition than is typically found in East or West Africa. Furthermore, this is an individually randomised trial that will monitor outcomes for an extended period, 4–5 months, after nutritional recovery. It also includes a group of children rarely studied but common in everyday practice: MAM children who do not receive supplemental food. This study design should allow for robust conclusions not only regarding the non-inferiority of OptiMA compared with a typical standard SAM protocol but also an evaluation of the incidence of SAM among children with MAM. This trial will also document outcomes of sustained health and nutrition status over a longer period than previous trials.

Furthermore, the stratified randomisation of children with SAM according to the WHO definition will shed some light on the clinical effects of the OptiMA strategy for this specific category of children. This inclusion strategy will also help to determine the segment of children with SAM defined by only WHZ with MUAC ≥125 mm, in Kasaï province where stunting prevalence in children is higher than anywhere else in the DRC.[36]

### Trial status

The recruitment phase began in July 2019 and ended in 22 January 2020. The estimated completion date for this study is September 2020.

**Author affiliations**
[1] University of Bordeaux, Inserm, French National Research Institute for Sustainable Development (IRD), Bordeaux Population Health Research Center, Team IDLIC, UMR 1219, Bordeaux, France
[2] The Alliance for International Medical Action (ALIMA), Paris, France
[3] The Alliance for International Medical Action (ALIMA), Kamuesha, Democratic Republic of Congo
[4] National Nutrition Programme (PRONANUT), Ministry of Health, Kinshasa, Democratic Republic of Congo
[5] Kamuesha Health Zone in the Kasaï Province, Ministry of Health, Kamuesha, Democratic Republic of Congo
[6] PACCI Research Programme, University Hospital of Treichville, Abidjan, Côte d'Ivoire
[7] The Alliance for International Medical Action (ALIMA), Dakar, Senegal

**Acknowledgements** We are indebted to the women and children who participate in the study. The authors wish to particularly acknowledge the ALIMA operational team on the ground who managed the study and the nutrition programme on a day-to-day basis. The authors also acknowledge the MoH of the DRC through the commitment of key representatives from the Kamuesha Health Zone and the National Nutrition Programme. We acknowledge the implementing partners of the interagency Nutrition Cluster Kasaï Province, in particular representatives from UNICEF and the World Food Programme. We also acknowledge Maguy Daures (Inserm UMR 1219, Bordeaux), Valérie Journot (Inserm UMR 1219, Bordeaux) and Claire Levy-Marchal (ALIMA, Paris) for having advising the study team on the trial methodology. This study was developed as part of the Clinical and Operational Research Alliance (CORAL). This research platform aims at developing high-quality innovative and transformative research programmes within a partnership between scientists from the research institute Inserm (Institut National de la Santé et la Recherche Médicale, Bordeaux and Abidjan) and the humanitarian organisation ALIMA (Alliance for International Medical Action), primarily on improving maternal and child health outcomes. A board of directors defines the scientific policy of the CORAL partnership alongside with the supervision of research projects and dissemination of results. It consists of senior representatives from both ALIMA and Inserm: Renaud Becquet (Inserm Bordeaux, methodological co-chair), Susan Shepherd (ALIMA, medical co-chair), Augustin Augier (ALIMA), Moumouni Kinda (ALIMA), Marie Jaspard (ALIMA and Inserm Abidjan), Claire Levy-Marchal (ALIMA) and Xavier Anglaret (Inserm Bordeaux and Abidjan). The CORAL research platform meets annually with an external scientific advisory board to review projects and strategic orientation. Finally, we warmly acknowledge the members of the Data Safety Monitoring Board (DSMB) for the OptiMA-DRC trial: Yves Martin-Prevel (DSMB chair, French National Institute for Sustainable Development (IRD), NUTRIPASS team, University of Montpellier, Montpellier, France), Matthew Colderon (Epicentre, New York, USA) and Katia Castetbon (DSMB statistician, School of Public Health, Université Libre de Bruxelles, Bruxelles, Belgium).

**Contributors** SS and RB developed the clinical (SS) and methodological (RB) study concept. RB, CC, XA, DG, AA, KP and SS designed the study methodology and wrote the protocol. CC, VH, HB, RA and MK coordinate the study teams. LIB, GTS coordinate the MoH staff working in the trial. CC, VH, HB, LIB, GTS, AB, CY and RB organise and supervise data collection. CY created the software tool for randomisation and developed the database. CC, VH, DG, XA, SS and RB developed the statistical analysis strategy. CC wrote the first draft of the manuscript. SS and

RB were primarily responsible for the final content of the manuscript. All authors critically reviewed the first draft and had a substantial writing contribution to the development of the final manuscript.

**Funding** The study was supported by the Innocent Foundation (London, UK), grant number ALIMA-2018-DRC, and received additional funding from European Commission through the European Civil Protection and Humanitarian Aid Operations (ECHO, Brussels, Belgium), grant number ECHO/COD/BUD/2018/91023. The funders had no role in study design, data collection and analysis, decision to publish or preparation of the manuscript. This document covers humanitarian aid activities implemented with the financial assistance of the European Union. The views expressed herein should not be taken, in any way, to reflect the official opinion of the European Union, and the European Commission is not responsible for any use that can be made of the information it contains.

**Competing interests** KP serves on the Social Purposes Advisory Commission of Nutriset, a main producer of lipid-based nutrient supplement products.

**Patient consent for publication** Not required.

**Provenance and peer review** Not commissioned; externally peer reviewed.

**Open access** This is an open access article distributed in accordance with the Creative Commons Attribution 4.0 Unported (CC BY 4.0) license, which permits others to copy, redistribute, remix, transform and build upon this work for any purpose, provided the original work is properly cited, a link to the licence is given, and indication of whether changes were made. See: https://creativecommons.org/licenses/by/4.0/.

**ORCID iDs**
Cécile Cazes http://orcid.org/0000-0002-5501-6224
Kevin Phelan http://orcid.org/0000-0002-7017-7464
Delphine Gabillard http://orcid.org/0000-0002-9122-2674
Xavier Anglaret http://orcid.org/0000-0003-3319-8423
Susan Shepherd http://orcid.org/0000-0001-9515-4333
Renaud Becquet http://orcid.org/0000-0003-3277-0985

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
