## [Reviewer comments · BMJ Open]

ARTICLE DETAILS

TITLE (PROVISIONAL)	Simplifying and optimizing management of acute malnutrition in children aged 6 to 59 months: study protocol for a community-based individually randomized-controlled trial in Kasai, Democratic Republic of Congo
AUTHORS	CAZES, Cécile; PHELAN, Kevin; Hubert, Victoire; ALITANOU, Rodrigue; BOUBACAR, Harouna; IZIE BOZAMA, Liévin; TSHIBANGU SAKUBU, Gilbert; BEUSCART, Aurélie; YAO, Cyrille; Gabillard, Delphine; KINDA, Moumouni; AUGIER, Augustin; ANGLARET, Xavier; Shepherd, Susan; Becquet, Renaud

VERSION 1 – REVIEW

REVIEWER	David Brewster Darwin, Australia
REVIEW RETURNED	07-Jul-2020

GENERAL COMMENTS	This is a well designed study which addresses an important issue. Carrying out this trial in the DRC context is vitally important. I only have a few questions to raise as the study is already well underway. 1. Has the research been affected by COVID-19? My concern is more about the researchers than the subjects.2. There is no mention of HIV. I think the protocol should say something about it, as it affects both growth and survival over the study period.3. Even excluding the additional material, I think the paper is too long and detailed, so would benefit from shortening.4. Although Table 1 indicates changes in MUAC as the main outcome, its use as such has not been well validated in studies. However, I see later in the text that weight gain is also being recorded, which may help validate using changes in MUAC in further studies or not.5. I am not keen on protocol papers, but agree that this protocol warrants publication due to its strength and originality. I am very pleased that it has been funded and is being implemented.
--

REVIEWER	David I Campbell Paediatric Gastroenterology Sheffield Children's Hospital UK
REVIEW RETURNED	16-Aug-2020

GENERAL COMMENTS	Thank you for submitting this manuscript, which is a research protocol for a very important study that is timely, important and relevant.
---

	Can I suggest that further proof readings of the english language are done. The ability to understand what is a very important study would be improved by reconstructing many of the sentences. Reading and re-reading the text gives the meaning, but makes it hard work. I was delighted to see a non-inferiority study against standard therapies, but giving the advantage of removing unhelpful subdivision in malnutrition subtypes and hence forms of nutritional rehabilitation. Could you give a reference for a negative appetite test?
--	--

VERSION 1 – AUTHOR RESPONSE

□ Reviewer: 1

1. Has the research been affected by COVID-19? My concern is more about the researchers than the subjects.

Since the beginning of COVID-19 pandemic, we reinforced hygiene procedures at each study site and during the follow-up visits realised in the village: systematic disinfection of MUAC bracelets, height and weight scales. Research officers and medical staff wore facial masks during consultations.

COVID-19 did not affect researchers in conducting the study according to the timeline and the protocol. The first case of COVID-19 was declared in Kasai province in July but not in the district where the trial took place.

2. There is no mention of HIV. I think the protocol should say something about it, as it affects both growth and survival over the study period.

Line 271-272: the reviewer is right and this point has been added in the manuscript (Intervention section paragraph, page 12).

3. Even excluding the additional material, I think the paper is too long and detailed, so would benefit from shortening.

We thank the reviewer for this comment. We removed some redundant details. The total number words of the manuscript is below 4000 as recommended by the editor for protocol publication.

4. Although Table 1 indicates changes in MUAC as the main outcome, its use as such has not been well validated in studies. However, I see later in the text that weight gain is also being recorded, which may help validate using changes in MUAC in further studies or not.

Yes, we will be able to compare the collinearity between MUAC and weight progression between the two strategies over a 6-month period.

5. I am not keen on protocol papers, but agree that this protocol warrants publication due to its strength and originality. I am very pleased that it has been funded and is being implemented.

We thank the reviewers for this comment.

□ Reviewer: 2

Thank you for submitting this manuscript, which is a research protocol for a very important study that is timely, important and relevant. Can I suggest that further proof readings of the English language are done. The ability to understand what is a very important study would be improved by reconstructing many of the sentences. Reading and re-reading the text gives the meaning, but makes it hard work.

Two of the co-authors (K. Phelan & S. Shepherd) are native English speakers and have extensively revised the manuscript in that way.

I was delighted to see a non-inferiority study against standard therapies, but giving the advantage of removing unhelpful subdivision in malnutrition subtypes and hence forms of nutritional rehabilitation. Could you give a reference for a negative appetite test?

The formulation was misleading indeed. We have therefore changed the term "negative appetite test" by the term "no appetite" (Line 179).